# Earliest known Gondwanan bird tracks: Wonthaggi Formation (Early Cretaceous), Victoria, Australia

Anthony J. Martin[1]*, Melissa Lowery[2], Michael Hall[2], Patricia Vickers-Rich[2], Thomas H. Rich[3], Claudia I. Serrano-Brañas[4,5], Peter Swinkels[3]

1 Department of Environmental Sciences, Emory University, Atlanta, Georgia, United States of America, 2 School of Earth, Atmosphere, and Environment, Monash University, Clayton, Victoria, Australia, 3 Museum Victoria, Melbourne, Victoria, Australia, 4 Benemérita Escuela Normal de Coahuila, Saltillo, Coahuila, Mexico, 5 Department of Paleobiology, National Museum of Natural History, Smithsonian Institution, Washington, DC, United States of America

☯ These authors contributed equally to this work.
* geoam@emory.edu

**Data Availability Statement:** All relevant data are within the paper and its Supporting information files.

## Abstract

The fossil record for Cretaceous birds in Australia has been limited to rare skeletal material, feathers, and two tracks, a paucity shared with other Gondwanan landmasses. Hence the recent discovery of 27 avian footprints and other traces in the Early Cretaceous (Barremian-Aptian, 128–120 *Ma*) Wonthaggi Formation of Victoria, Australia amends their previous rarity there, while also confirming the earliest known presence of birds in Australia and the rest of Gondwana. The avian identity of these tracks is verified by their tridactyl forms, thin digits relative to track lengths, wide divarication angles, and sharp claws; three tracks also have hallux imprints. Track forms and sizes indicate a variety of birds as tracemakers, with some among the largest reported from the Early Cretaceous. Although continuous trackways are absent, close spacing and similar alignments of tracks on some bedding planes suggest gregariousness. The occurrence of this avian trace-fossil assemblage in circumpolar fluvial-floodplain facies further implies seasonal behavior, with trackmakers likely leaving their traces on floodplain surfaces during post-thaw summers.

## Introduction

Present-day Australia hosts a remarkable diversity of birds, with their evolutionary lineages well represented in the Cenozoic fossil record there [1–4]. In contrast, evidence of Mesozoic birds in Australia is scanty, denoted by isolated skeletal elements, a few feathers, and two footprints in Cretaceous strata [5–11]. The oldest definite body fossil evidence for birds in Australia consists of a furcula and a flight feather from the Early Cretaceous (Valangian-Aptian) Wonthaggi Formation of Victoria [8, 11]. As for the rest of Gondwana, the only Early Cretaceous bird outside of Australia is *Cratoavis cearensis* from the Crato Formation (Aptian) of Brazil [12, 13], with bird tracks reported from the Late Cretaceous of Argentina [14–17] and Tunisia [18]. Given so few Mesozoic avian fossils, then, we have little evidence of when birds

**Funding:** The author(s) received no specific funding for this work.

**Competing interests:** No authors have competing interests.

got their start in Australia and most other Gondwanan landmasses, let alone how they interacted with their environments.

Hence the recent discovery of 27 bird tracks and additional traces in Barremian-Aptian (128–120 *Ma*) circumpolar fluvial facies of the Wonthaggi Formation of Victoria, Australia represents a significant advance in the fossil record of Early Cretaceous birds in Australia and the remainder of Gondwana. These trace fossils not only affirm an avian presence at this time and place, but the variety of sizes and track forms also imply a diverse avifauna in formerly polar environments of Australia [8–10, 19, 20]. Moreover, some specimens are among the largest documented bird footprints from the Early Cretaceous, rivaling sizeable avian tracks from east Asia [21]. Although outcrop expressions of the tracks are limited and no continuous trackways are evident, close spacing and similar orientations of tracks on some bedding planes further point to gregariousness. This evidence, as well as the sedimentological conditions of circumpolar fluvial-floodplain facies, suggest seasonally linked behaviors and preservation of their tracks during polar summers, while also providing glimpses of the early history, biodiversity, and adaptations of birds in Gondwana.

## Materials and methods

Nearly all tracks described in this study were discovered by one of us (ML) in 2020–2022 while prospecting for fossils on Bass Coast marine platforms near Inverloch, Victoria. The remainder of tracks were noticed in May 2022 during a site visit by another of us (AJM) in the presence of most coauthors. Of the 27 tracks documented in this study, 26 were at a locality dubbed "Footprint Flats," and one was 1.2 km southwest of there, at a place nicknamed "Honey Bay"; the Wonthaggi furcula [8] was recovered from Flat Rocks, also known as "Dinosaur Dreaming" (Fig 1). For the track at Honey Bay, it was simply labeled HB-1, whereas tracks at Footprint Flats were assigned specimen numbers for each of five beds in stratigraphic order from oldest to youngest, with FF-1-1 on the lowest bed through FF-5B-5 on the uppermost bed (Fig 2). The top two assemblages at Footprint Flats are in a silty sandstone bed but on different bedding planes, hence tracks in this bed were assigned to two levels, 5A (lower) and 5B (upper). Lithology, physical sedimentary structures, and additional biogenic sedimentary structures (e.g., invertebrate burrows) associated with the tracks were also noted and evaluated. Beds on marine platforms at both localities were dipping 10–15˚ east-northeast, resulting in narrow exposures of bedding planes that likely prevented our observing continuous trackways. With no sure way to tell whether or not tracks were from the same trackmakers, all were treated as individual specimens.

Tracks described in this study were measured with Mitutoyo™ digital calipers and track measurements were recorded to the nearest millimeter. Measured parameters included track length, track width, digit lengths and widths, and interdigital angles between digits I-II (if digit I was evident), digits II-III, and digits III-IV (Fig 3). Track length was measured from the rearward-most point where the midlines of digits II-IV met on an axis defined by digit III, and track width was measured from the outermost edges of digits II and IV. Extended track length was also noted for tracks with traces of digit I. Because nearly all tracks were preserved as negative-relief features on bed tops, measurements were taken as minimum outlines from where track bottoms ("floors") met their sides ("walls"), rather than where they intersected uppermost bedding planes. Length:width ratios were calculated from length and width measurements, and divarication was the sum of interdigital angles II-III and III-IV. Track orientations were measured as azimuths with a Silva™ compass placed above the linear trend of digit III on each track.

Descriptive statistics–such as mean and standard deviation–were then applied to tracks expressed as tridactyl patterns (digits II-IV). As for qualitative data, we noted presence or

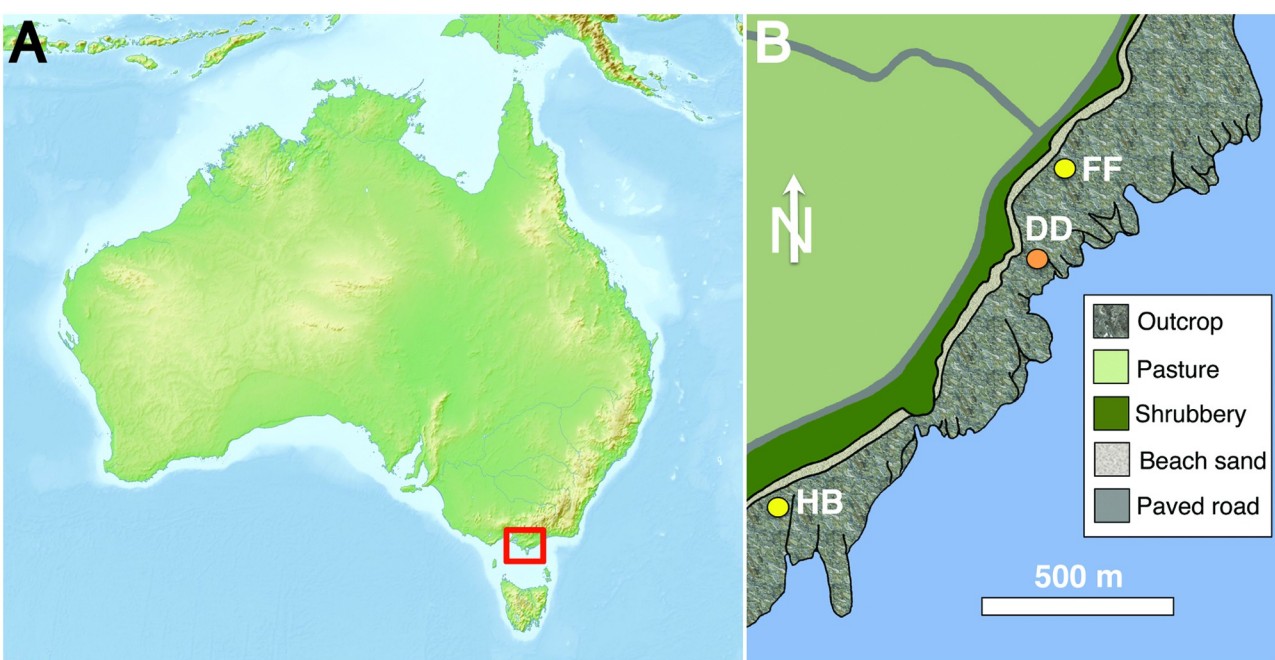

**Fig 1. Locality map of Wonthaggi Formation avian tracksites, Victoria, Australia.** (A)Approximate outcrop area of Wonthaggi Formation indicated (red box). Map of Australia downloaded from Mapswire (https://mapswire.com/maps/australia/), which are provided under a Creative Commons (CC-BY 4.0) license; accessed and retrieved September 21, 2023. (B) Wonthaggi Formation coastal outcrops and locations of avian tracksites for this study, with "Honey Bay" (HB) at S 38˚ 39.9', E 145˚ 40.5' and "Footprint Flats" (FF) at S 38˚ 39.4', E 145˚ 41.0'; "Dinosaur Dreaming" (DD) site is at S 38˚ 39.5", E 145˚ 41.2'. Map was drawn and adapted from public-domain satellite images at EarthExplorer (https://earthexplorer.usgs.gov).

absence of phalangeal pads, ungual (claw) impressions, interdigital webbing, and sedimentary structures outside the footprint outlines formed by tracemaker interactions with the host sediment [9, 22–24]. Orientations of claw impressions on digits II and IV relative to the track midlines were also noted, as these are often used as a diagnostic criterion for bird tracks [25–28]. One of us (AJM) made detailed line drawings of tracks and closely associated sedimentary structures (e.g., bedding, burrows), with outline sketches of the inferred foot anatomy overlain for each footprint.

Tracks were exposed on an active marine platform, and hence have been subjected to physical weathering from daily tidal and wave activity, as well as bioerosion and encrustation by algae, barnacles, and molluscans. Most tracks examined in this study were only exposed at low tide, attesting to conditions that assured ephemeral preservation. For example, in one spot, seven tracks were erased from a bedding plane at Footprint Flats (FF-5) over the course of 18 months, from November 2020 to May 2022. Given these conditions, one of us (PS) photographed and recorded GPS coordinates of track-bearing bedding planes, then made silicone molds and polyester-resin casts, which will serve as lasting and tactile (non-digital) records of track forms and dimensions (S1 File). For tracks eroded since their initial discovery, we measured track parameters from their polyester-resin casts, with those specimens indicated (Table 1). In May 2022, we also placed a cast of seven tracks from bed FF-5 onto their former bedding planes to measure in-situ orientations of those footprints on the two surfaces of that bed (levels 5A and 5B; Fig 4).

Owing to safety protocols in Australia necessitated by the Covid-19 pandemic throughout 2020–2022, we limited the number of participants and time together in the field while documenting tracks and their geological context on site. This documentation included field

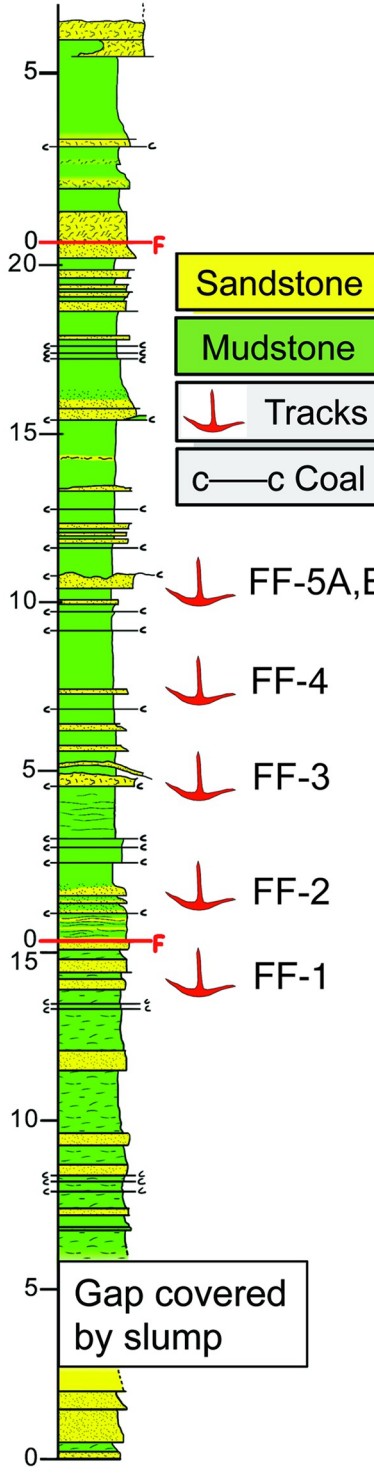

**Fig 2. Stratigraphic section of Wonthaggi Formation at Footprint Flats locality.** Sandstone bed numbers (FF-1 through FF-5A-B) for stratigraphic levels bearing tracks. "F" and red lines denote faults, with one fault less than a meter above FF-1 and another about 11 meters above FF-5. Colors used in this section do not match those of lithofacies.

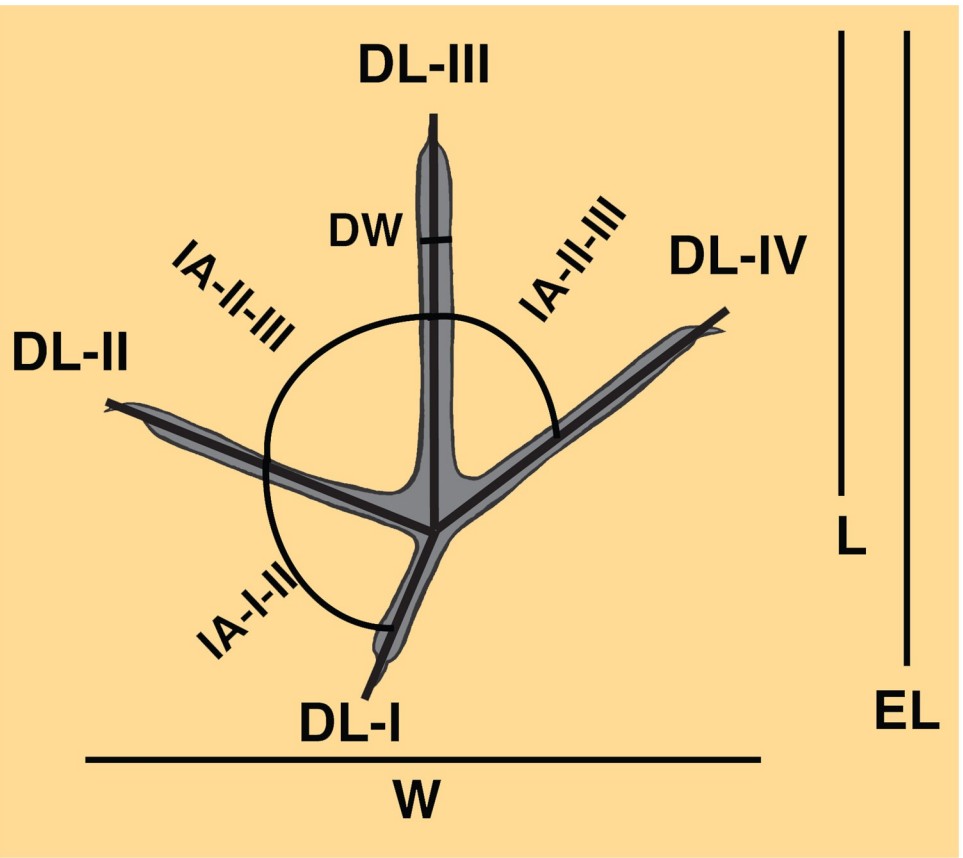

**Fig 3. Measurements applied to avian tracks in this study.** L = length, W = width, EL = extended length (with digit I), DL = digit length, DW = digit width, IA-I-II = interdigital angle for digits I-II (if present), IA-II-III = interdigital angle for II-III, IA-III-IV = interdigital angle for III-IV. Total divarication is the sum of interdigital angles for digits II-IV. Example track is depicted with average interdigital angles for all tracks with digits II-IV in this study (n = 25), with digits II-III = 53˚, digits III-IV = 63˚, and a divarication of 116˚.

mapping by one of us (MH), and others taking two-dimensional digital photos, measurements, and descriptions, and making labeled field sketches. As mentioned previously, one of us (AJM) later made line drawings based on photographs taken from 2020–2022. Although we did not apply photogrammetric methods recommended for vertebrate footprints [29], we are nonetheless confident that our non-photogrammetric methods were more than adequate for accurately describing and interpreting the Wonthaggi Formation footprints. If future researchers wish to test and replicate our results, we encourage them to examine polyester-resin casts of the tracks, which will be curated by Museum Victoria. These casts then can be further studied through photogrammetry, laser scanning, and other digital methods.

## Results

### Study area and stratigraphy

All tracks described in this study were exposed in marine-platform outcrops of the Wonthaggi Formation west of Inverloch, Victoria. As mentioned before, tracks at Footprint Flats were preserved on five stratigraphic levels (FF-1 to FF-5) on tops of ripple-bedded and silty to fine- to medium-grained sandstones; these sandstones were interbedded with mudstones and thin

**Table 1. Measurements and descriptive statistics of Wonthaggi Formation tracks at Honey Bay (n = 1) and Footprint Flats (n = 26).**

(A) Anisodactyl incumbent tracks (digits II-IV) from Footprint Flats (FF) and Honey Bay (HB), Wonthaggi Formation, Victoria, Australia.

| Track | L | W | L/W | D2-L | D2-W | D3-L | D3-W | D4-L | D4-W | D23-A | D34-A | TD | Az |
|---|---|---|---|---|---|---|---|---|---|---|---|---|---|
| FF-1-1 | 44 | 75 | 0.59 | 34 | 5 | 46 | 6 | 43 | 5 | 55 | 70 | 125 | 255 |
| FF-1-2 | 45 | 65 | 0.69 | 39 | ND | 45 | 4 | 41 | 5 | 55 | 64 | 119 | 190 |
| FF-1-4 | 64 | 97 | 0.66 | 57 | 4 | 64 | 5 | 58 | 4 | 53 | 73 | 126 | 210 |
| FF-2-1 | 58 | 89 | 0.64 | 53 | 5 | 58 | 6 | 57 | 5 | 53 | 58 | 111 | 170 |
| FF-3-1 | 56 | 78 | 0.72 | 44 | 1 | 56 | 11 | 48 | 13 | 58 | 64 | 122 | 170 |
| FF-3-2 | 88 | 92 | 0.96 | 58 | 9 | 88 | 11 | 58 | 12 | 58 | 67 | 125 | 340 |
| FF-3-3 | 85 | 96 | 0.89 | 57 | 7 | 85 | 12 | 62 | 15 | 53 | 60 | 113 | 290 |
| FF-3-5 | 53 | 72 | 0.74 | 41 | 7 | 53 | 9 | 41 | 8 | 50 | 69 | 119 | 315 |
| FF-3-6 | 117 | 132 | 0.89 | 67 | 9 | 117 | 1 | 74 | 6 | 55 | 64 | 119 | 55 |
| FF-4-1 | 59 | 76 | 0.78 | 43 | 7 | 59 | 6 | 55 | 5 | 64 | 65 | 129 | 155 |
| FF-4-2 | 77 | 98 | 0.79 | 62 | 3 | 77 | 6 | 57 | 6 | 52 | 60 | 112 | 340 |
| FF-4-3 | 102 | 142 | 0.72 | 90 | 11 | 102 | 9 | 84 | 7 | 52 | 55 | 107 | 160 |
| FF-4-4 | 92 | 116 | 0.79 | 75 | 1 | 92 | 9 | 77 | 16 | 53 | 60 | 113 | 165 |
| FF-4-5 | 91 | 122 | 0.75 | 78 | 14 | 91 | 11 | 86 | 14 | 57 | 60 | 117 | 330 |
| FF-4-6 | 102 | 121 | 0.84 | 59 | 4 | 102 | 11 | 70 | 6 | 52 | 56 | 108 | 135 |
| FF-5A-1 | 55 | 84 | 0.65 | 47 | 3 | 55 | 3 | 49 | 5 | 52 | 64 | 116 | 230 |
| FF-5A-2 | 61 | 82 | 0.74 | 41 | 4 | 61 | 4 | 51 | 3 | 46 | 55 | 101 | 230 |
| FF-5A-3 | 79 | 84 | 0.94 | 53 | 5 | 79 | 5 | 69 | nd | 54 | 61 | 115 | 235 |
| FF-5A-4 | 63 | 80 | 0.79 | 45 | 3 | 63 | 5 | 45 | 3 | 55 | 64 | 119 | 130 |
| FF-5A-5 | 69 | 78 | 0.88 | 44 | 4 | 69 | 4 | 35 | 4 | 49 | 52 | 101 | 270 |
| FF-5B-1 | 69 | 76 | 0.91 | 42 | 4 | 69 | 6 | 44 | 3 | 45 | 61 | 106 | 140 |
| FF-5B-3 | 91 | 116 | 0.78 | 69 | 5 | 91 | 6 | 54 | 6 | 48 | 64 | 112 | 105 |
| HB-1 | 46 | 77 | 0.6 | 34 | 5 | 46 | 6 | 43 | 5 | 67 | 71 | 138 | 240 |
| Mean | 72 | 93 | 0.77 | 54 | 5 | 73 | 7 | 57 | 5 | 67 | 71 | 138 | 211 |
| Standard Deviation | 20 | 21 | 0.11 | 54 | 3 | 20 | 3 | 14 | 4 | 5 | 5 | 9 | 79 |

(B) Anisodactyl tracks (digits I-IV) from Footprint Flats, Wonthaggi Formation, Victoria, Australia.

| Track | L | W | EL | L/W | D1-L | D1-W | D2-L | D2-W | D3-L | D3-W | D4-L | D4-W | D12-A | D23-A | D34-A | TD | Az |
|---|---|---|---|---|---|---|---|---|---|---|---|---|---|---|---|---|---|
| FF-1-3 | 76 | 96 | 86 | 0.79 | 18 | 4 | 61 | 5 | 76 | 7 | 74 | 5 | 127 | 48 | 56 | 104 | 265 |
| FF-5B-2 | 83 | 118 | 98 | 0.7 | 19 | 4 | 48 | 6 | 83 | 3 | 67 | 6 | 88 | 54 | 69 | 123 | 150 |
| FF-5B-4 | 93 | 115 | 102 | 0.81 | 21 | 5 | 43 | 6 | 93 | 8 | 65 | 8 | 92 | 55 | 69 | 124 | 160 |
| Mean | 84 | 110 | 95 | 0.77 | 19 | 4 | 51 | 6 | 84 | 6 | 69 | 6 | 102 | 52 | 65 | 117 | 192 |
| Standard Deviation | 9 | 12 | 8 | 0.06 | 2 | 1 | 9 | 1 | 9 | 3 | 5 | 2 | 21 | 4 | 8 | 11 | 64 |

(C) Incomplete anisodactyl incumbent tracks (digit III and one lateral digit) from Footprint Flats, Wonthaggi Formation, Victoria, Australia

| Track | L | (W) | L/W | D2-L | D2-W | D3-L | D3-W | D23-A | Az |
|---|---|---|---|---|---|---|---|---|---|
| FF-3-4 | 82 | 94* | 0.87* | 43 | 9 | 82 | 14 | 54 | 275 |
| FF-5B-5 | 66 | 69* | 0.96* | 61 | 5 | 66 | 7 | 47 | 140 |

(D) Summary descriptive statistics for anisodactyl incumbent and anisodactyl tracks with digits II-IV (n = 25).

| | L | W | L/W | D2-L | D2-W | D3-L | D3-W | D4-L | D4-W | D23-A | D34-A | TD | Az |
|---|---|---|---|---|---|---|---|---|---|---|---|---|---|
| Mean | 74 | 95 | 0.77 | 53 | 5 | 74 | 7 | 58 | 7 | 54 | 63 | 116 | 209 |

(*Continued*)

| Standard Deviation | 20 | 21 | 0.1 | 14 | 3 | 20 | 3 | 14 | 4 | 5 | 6 | 9 | 76 |
|---|---|---|---|---|---|---|---|---|---|---|---|---|---|

(A) Track = specimen number; L = length (mm); W = width (mm); L/W = length:width ratio; D2-L = digit II length; D2-W = digit II width; D3-L = digit III length; D3-W = digit IV width; D4-L = digit IV length; D4-W = digit IV width; D23-A = angle between digits II and III (degrees); D34-A = angle between digits III and 4; TD = total divarication (sum of D23-A and D34-A); Az = azimuth (compass degrees); nd = not determinable.

(B) Track = specimen number; L = length (mm); W = width (mm); EL = extended length (with digit I); L/W = length:width ratio; D1L = digit I length; D1W = digit I width; D2-L = digit II length; D2-W = digit II width; D3-L = digit III length; D3-W = digit IV width; D4-L = digit IV length; D4-W = digit IV width; D12-A = angle between digits I and II; D23-A = angle between digits II and III (degrees); D34-A = angle between digits III and 4; TD = total divarication (sum of D23-A and D34-A); Az = azimuth (compass degrees).

(C) L = length (mm); (W) = extrapolated width (based on 2X digit 3 + lateral digit);

(D) L = length (mm); W = width (mm); L/W = length:width ratio; D2-L = digit II length; D2-W = digit II width; D3-L = digit III length; D3-W = digit IV width; D4-L = digit IV length; D4-W = digit IV width; D23-A = angle between digits II and III (degrees); D34-A = angle between digits III and 4; TD = total divarication (sum of D23-A and D34-A); Az = azimuth (compass degrees).

(< 1 cm thick) coal beds in a 7-m interval (Fig 2). The single track at Honey Bay was similarly on the top surface of a ripple-bedded fine-grained sandstone. Based on palynological data from the Wonthaggi Formation in this area, track-bearing strata at both localities are interpreted as late Barremian to early Aptian (128–120 *Ma*) [10, 30]. Two faults interrupt the

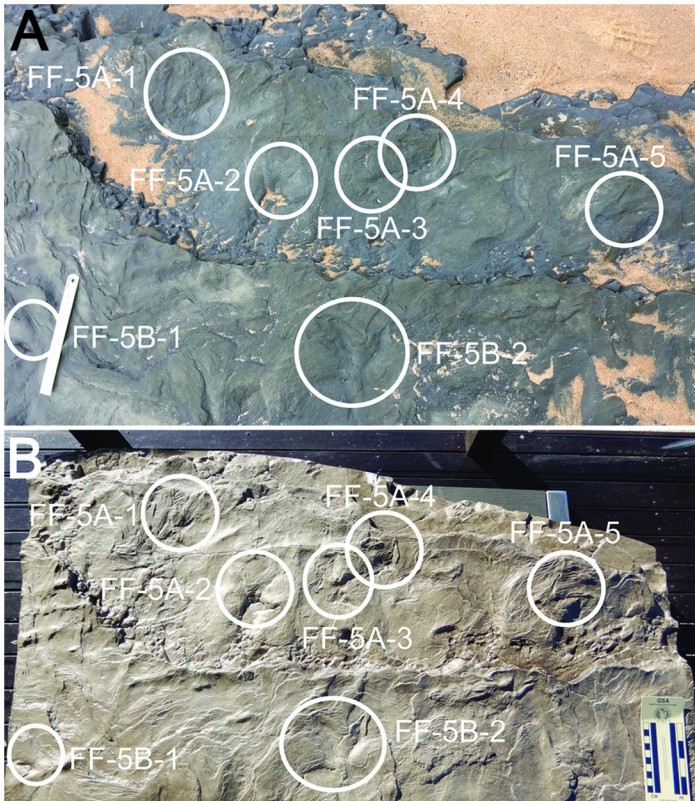

**Fig 4. Polyester-resin cast duplicating tracks in rocky intertidal environment at Footprint Flats (FF) locality.** (A) Bedding planes of FF-5A (lower surface) and FF-5B (upper surface), with seven tracks (circled) photographed just before molding on November 11, 2020; ruler = 15 cm long. (B) Resin cast of bedding plane, with same seven tracks indicated; scale with centimeters. See text for key to specimen numbers, Table 1 for measurements of tracks, and S1 File for description of molding and casting methods.

stratigraphic sequence at the Footprint Flats site, with FF-1 just below the one fault and the remaining four track-bearing strata below the second fault. However, we did not detect any repetition of strata at this locality as a result of faulting.

The Wonthaggi Formation is a thick sequence of clastic conglomerates, sandstones, and mudstones formed mostly by alluvial, fluvial, and lacustrine processes in rift valleys of the Gippsland Basin [31–34]. Based on a wide range of geological evidence, southern Australia was affected by glaciation during Early Cretaceous [35, 36]; the paleolatitude of the Gippsland basin was 75–80˚ S as southern Australia diverged from Antarctica then [19, 20, 33, 37–41]. Although mean annual air temperatures of polar regions during the Cretaceous were higher than those of modern settings, these ecosystems and their communities still experienced freezing temperatures and months of darkness [20, 39–44]. Ice accumulation during winter months led to torrential run-off and high depositional rates in polar springs, followed by waning flows and emergent floodplains and channel margins [32–34, 39]. Given these climatic and sedimentological conditions, most biogenic sedimentary structures–such as footprints and burrows–were more likely formed and preserved on floodplain surfaces exposed during lower flow regimes, such as during late springs and summers. Hydrological, sedimentological, and ichnological conditions of a modern fluvial point bar during summer in the Arctic Circle were used as analogs for the formation and preservation of the trace fossils reported here [24].

## Descriptions of lithofacies and tracks

Most tracks at Footprints Flats were evident as negative-relief epichnia, forming impressions on bed tops of 10–20 cm thick ripple-bedded sandy siltstones and fine-grained sandstones. However, a few footprints were also preserved as positive-relief epichnia, with their sediment fills more resistant to weathering compared to the host substrate. Tracks are preserved on at least five stratigraphic levels, with the highest on different bedding planes within the same stratum. The single track from Honey Bay (HB-1) was expressed in positive-relief (1–3 mm) on a 15-cm thick ripple-bedded fine-grained sandstone, but was also definable as a disturbance of underlying and surrounding bedding (Fig 5).

Using the terminology of Elbroch and Marks [45], tracks were categorized as anisodactyl (traces of digits I-IV present) or anisodactyl incumbent (only digits II-IV present). Tracks with only digit III and one lateral digit preserved were termed as incomplete. Of these categories, anisodactyl incumbent tracks are the most common (n = 22), followed by anisodactyl (n = 3), and incomplete (n = 2). Other than the 25 complete and two incomplete tracks, several single-digit and/or ungual impressions were also present on bedding planes, such as those near or cross-cut by more complete footprints (Fig 6). Although tracks represented by single-digit or ungual traces were not included in our total track count, these are still worth noting as additional evidence of trackmakers.

Most tracks from Footprint Flats are relatively shallow (1–8 millimeters), with some parts impressed deeply enough to intersect underlying mudstone layers, but other portions nearly coincided with uppermost bedding planes. As a result, few tracks expressed the full foot anatomy of their makers, with incomplete prints as the norm (Fig 7). Although skin impressions were not observed in any tracks, outlines of phalangeal pads were discernable in a few, such as FF-4-1, FF-4-3, and FF-4-5. Most digits taper distally along their lengths, and lateral digits typically end with thin and pointed ungual impressions that curve posteriorly. In some specimens, incomplete digits were only defined only by distal claw imprints and/or linear disruptions of bedding (Fig 8).

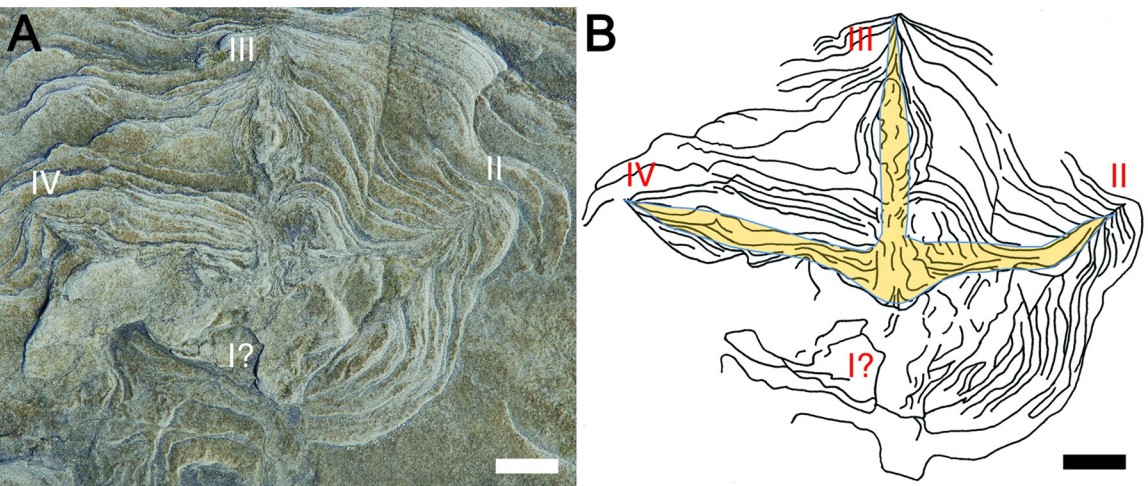

**Fig 5. Avian track in sandy siltstone from Wonthaggi Formation at Honey Bay locality.** (A) Footprint HB-1, with digits II-IV indicated. (B) Line drawing of HB-1 footprint with overlay of inferred foot morphology. Note deformation of bedding immediately behind digit III and the remainder of the track, suggesting the possible effect of a digit I (I?). Scale = 1 cm in both.

Proximal interdigital webbing (semipalmated) seemed present in a few tracks, such as FF-4-3, FF-4-4, and FF-4-5 (Figs 6 and 7), but these and other anatomical features were difficult to distinguish from sedimentary structures formed by an interaction of foot movement and original sedimentary conditions. For example, track FF-1-4 has a swath of disturbed

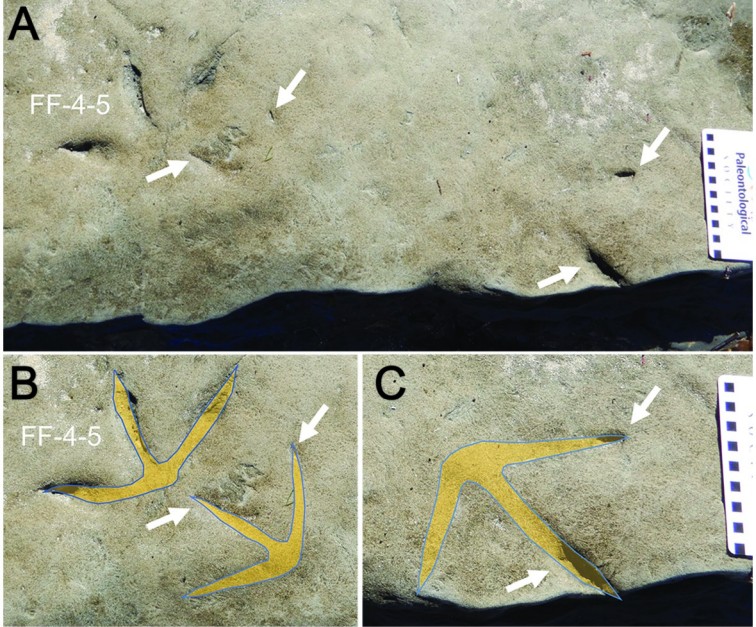

**Fig 6. Ungual and digit impressions near a more clearly defined avian footprint in the Wonthaggi Formation, Footprint Flats.** (A) Two ungual prints next to FF-4-5 (left) and a nearby incomplete track (two digits registered), with each ungual and digit denoted by arrows; ungual print behind FF-4-5 gives misleading appearance of a digit I for that track. (B) Overlay of inferred foot morphology for FF-4-5 and estimated anisodactyl-incumbent track based on two ungual prints. (C) Overlay of inferred foot morphology for incomplete track with only partial prints of two digits. For paired ungual and digit prints, one is presumed as digit III and the other as either II or IV. Photo scale = 5 millimeter squares.

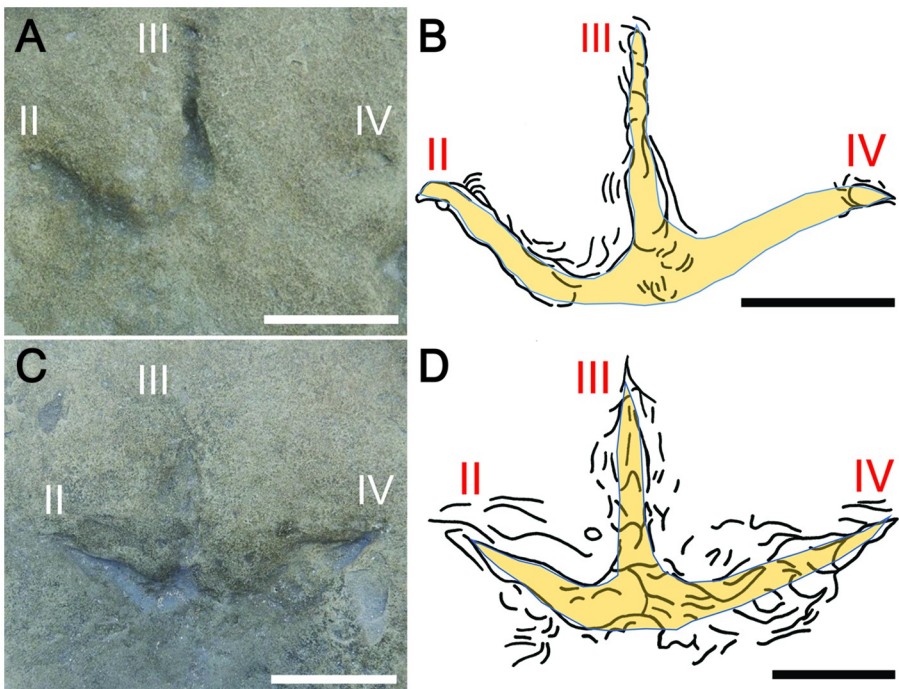

**Fig 7. Wonthaggi Formation avian tracks with uneven depths of digit impressions.** (A, B) Photo (A) and line drawing (B) of track FF-4-3, interpreted as a right pes, with overlay of inferred foot morphology. Digits II and III penetrated to the underlying mudstone layer, but with shallow and distal impressions of digit IV. (C, D) Photo (C) and line drawing (D) of FF-4-4, interpreted as a right pes with overlay of inferred foot morphology. Digits II and IV reached the underlying mudstone layer, but digit III barely registered. Proximal interdigital webbing is inferred for both tracks, but these features also may be from foot-sediment interactions. Scale = 5 cm in all parts; see Fig 2 and Table 1 for stratigraphic position and specimen measurements, respectively.

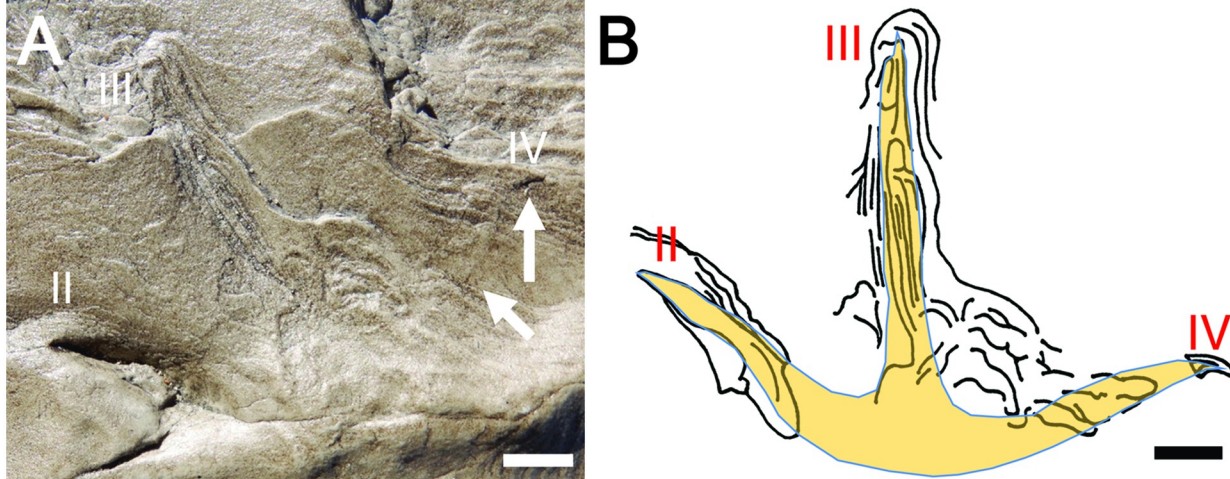

**Fig 8. Wonthaggi Formation avian track with digits defined by distal ends and ungual impressions.** (A) Photo and (B) line drawing of track FF-5B-1 (polyester cast), interpreted as right pes, with ungual imprint of digit IV and slight disruption of bedding (arrows) as evidence for that digit, and inferred foot morphology overlain. Scale in both = 1 cm.

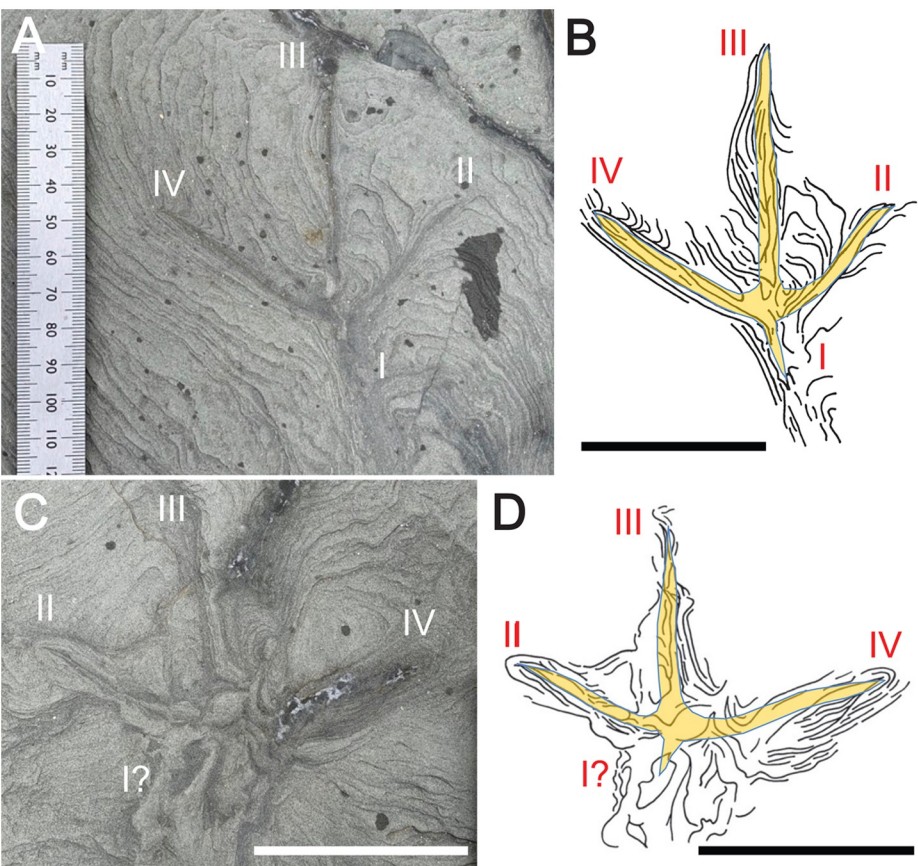

**Fig 9. Differing sedimentary expressions of digit I in Wonthaggi avian tracks.** (A, B) Photo and line drawing FF-1-3, with inferred foot morphology overlain. Because digit I is clearly defined, this track is classified as anisodactyl. (C, D) Photo and line drawing of FF-1-4. A probable digit I impression in FF-1-4 is obscured by disturbed sediment behind digits II-IV. Because of this uncertainty, the latter track was classified as anisodactyl incumbent. Scale = 5 cm in both B and D.

sediment behind digits II-IV, obscuring any outline of a probable digit I impression there (Fig 9).

Other extramorphological structures are associated with two of the three anisodactyl tracks, FF-5B-2 and FF-5B-4, both of which have prominent ridges in front of digits II-IV, as well as linear grooves and ridges outlining digit I (Figs 10 and 11). Two anisodactyl incumbent tracks that nearly intersect one another, FF-5A3 and FF-5A-4, as well as a nearby single-digit impression, also had noticeable ridges around track outlines (Fig 12). These sedimentary structures are likely related to foot movement interacting with sediments, explained more later.

In terms of descriptive statistics, average sizes and standard deviations for all tracks with digits II-IV expressed were 74 ± 20 mm long and 95 ± 21 mm wide, with ranges of 44 to 117 mm long and 65 to 142 mm wide (n = 25; Table 1, Fig 13). Digit II and digit IV trace lengths averaged 53 ± 14 mm and 58 ± 14 mm, respectively. Widths of digit traces varied from 3 to 14 mm, but traces of digits II, III, and IV averaged 5–7 mm wide, or less than 10% of footprint lengths. Length:width ratios of tracks ranged from 0.59 to 0.96, with a mean of 0.77. Interdigital angles for digits II-III ranged from 45˚ to 67˚, with a mean and standard deviation of 53.5 ± 4.8˚, whereas interdigital angles for digits III-IV were from 52˚ to 71˚, and with a mean

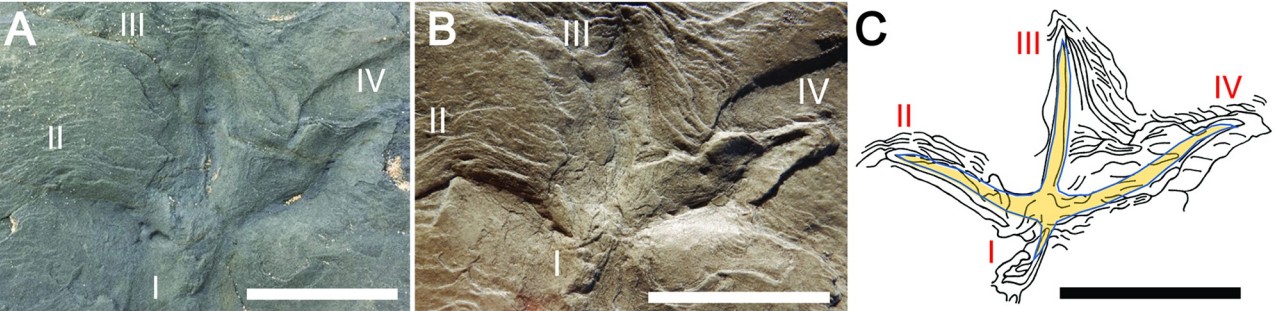

**Fig 10. Anisodactyl track FF-5B-2.** (A) Outcrop view of track on November 11, 2020. (B) Polyester-resin cast of track. (C) Line drawing with overlay of inferred foot morphology. Track is interpreted as a right pes with full anisodactyl expression, sedimentary structures (ridges) in front of digits II-IV, and a thin hooked groove associated with digit I. Scale bar = 5 cm in all parts.

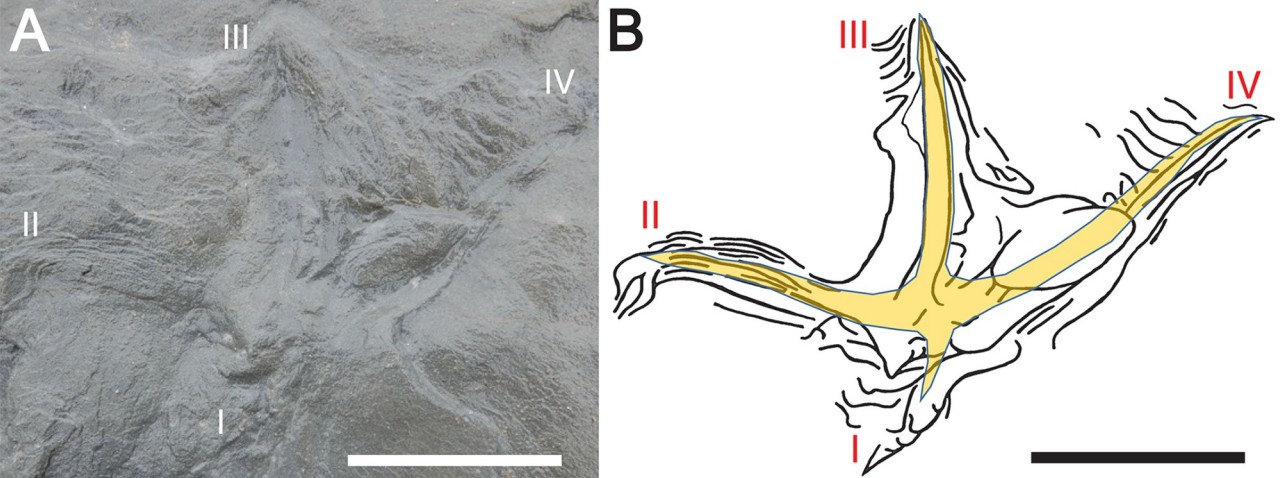

**Fig 11. Anisodactyl track FF-5B-4.** (A) Outcrop view of track on May 25, 2022. (B) Line drawing with overlay of inferred foot morphology. Track is interpreted as right pes, with sedimentary structures (ridges) in front of digits II-IV and a thin linear groove associated with digit I, presumably imparted by the ungual. Scale bar = 5 cm in both parts.

and standard deviation of 63 ± 6˚. Total divarication angles between digits II and IV ranged from 101˚ to 139˚ and averaged 116 ± 9˚.

However, sources of potential variations and errors in track measurements include their exposure on a marine platform, which resulted in considerable physical and biological weathering of some track features. For example, a few tracks on bed FF-3 had widened digits altered by erosion, whereas others had barnacles affixed to digit impressions and gastropods actively grazing algae on their surfaces (Fig 14).

Folding and faulting of Wonthaggi strata rendered track orientations meaningless in a regional sense, but these data were applicable to local clusters of tracks sharing the same bedding plane. For example, at Footprints Flats, four closely spaced tracks on FF-3 and a possible fifth track had southwest-to-northwest orientations between 235˚ and 340˚ (Fig 15A). On FF-5A, four of five closely spaced tracks were within a narrower range of orientations, from 230˚ to 270˚, whereas two tracks on stratigraphic level 5B were identically oriented at 165˚ (Fig 15B). These data help test for possible tracemakers' group behaviors, discussed later.

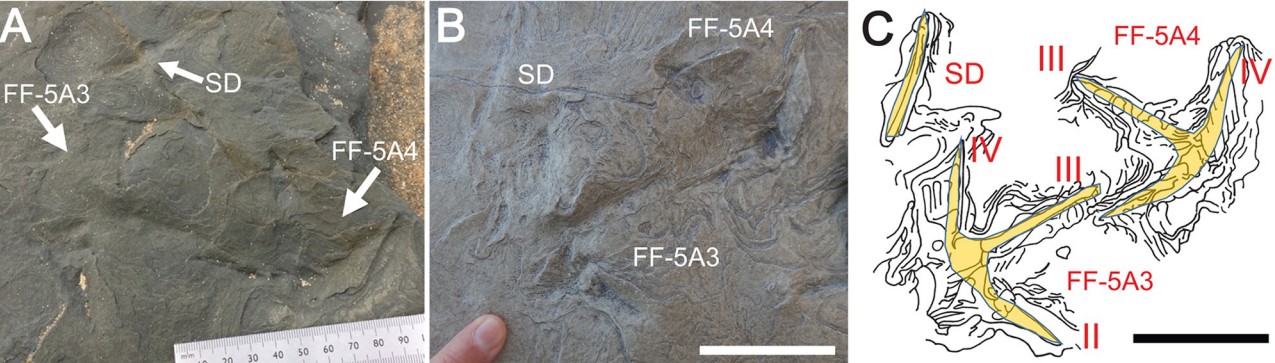

**Fig 12. Closely associated Wonthaggi tracks and single-digit impression with prominent extramorphological features.** (A) Outcrop view of FF-5A3, FF-5A-4, and single digit (SD); photo taken on November 11, 2020. (B) Polyester resin cast of tracks and single-digit impression. (C) Line drawings of FF-5A-3 and FF-5A-4 tracks, single-digit impression, and extramorphological structures, with overlays of inferred foot morphology for each track and digit. Ruler in A is in millimeters, scale bar in B and C = 5 cm.

Although the Wonthaggi tracks vary in sizes and forms, they share the following morphological traits: anisodactyl or anisodactyl incumbent, with three forward-pointing digits (II-IV); a limited size range (<15 cm length); relatively thin digits relative to footprint length (<10%); divarication angles between digits II and IV that consistently exceeded 90°, which corresponded with track length:width ratios of about 0.8; and narrow, pointed ungual (claw) impressions at digit ends. In most instances, digit II and IV ungual imprints also curved posteriorly with respect to the main axis of their digits (Figs 6–10). As mentioned previously, at least three tracks (FF-1-3, FF-5B-2, FF-5B4), and perhaps a fourth (FF-1-4) also had a digit I imprint, resulting in fully anisodactyl tracks. All other tracks with no clearly defined digit I were considered as anisodactyl incumbent. Proximal webbing may be present at intersections between traces of digit III and digit II or IV of some tracks, but none have evidence of distal webbing as fully palmate tracks [45].

Invertebrate trace fossils in track-bearing strata at Footprint Flats include small (5–10 mm wide) and scarce horizontal to inclined burrows intersecting rippled bed tops that occasionally

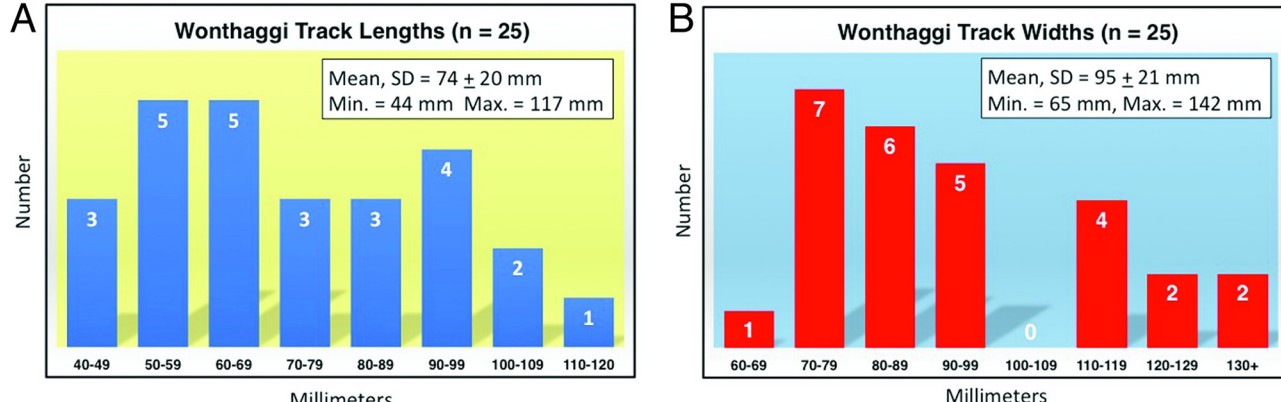

**Fig 13. Size-frequency distributions of lengths and widths for Wonthaggi Formation tracks with digits II-IV.** (A) Track lengths. (B) Track widths. Bins are 10 mm for each parameter (e.g., 40–49 mm), with number of tracks in each bin and mean, standard deviation, minimum, and maximum values; lengths exclude extended lengths for tracks with digit I. See Table 1 for specimen numbers and values.

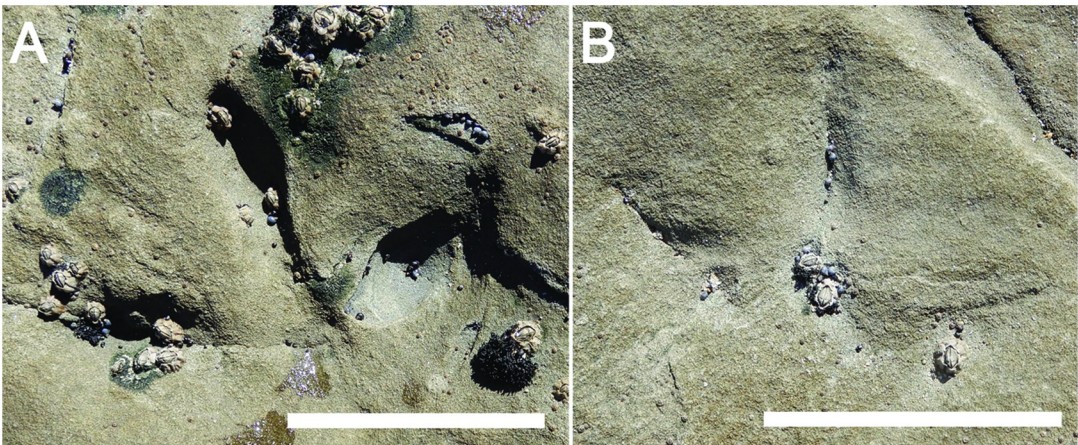

**Fig 14. Wonthaggi bird tracks affected by modern erosion and marine organisms at Footprint Flats locality.** (A) FF-3-1-1, interpreted as left pes, with widened (eroded) digits and encrusted by algae, barnacles, and gastropods. (B) FF-3-5, interpreted as right pes, with barnacles and gastropods on track center and within digits. Scale bar = 5 cm in both.

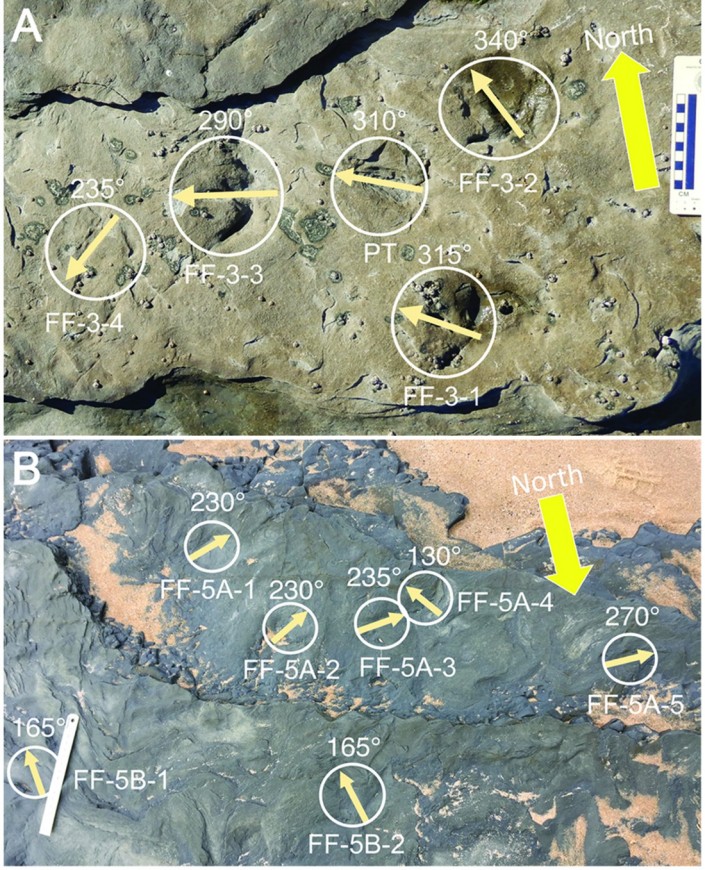

**Fig 15. Similar orientations of closely spaced bird-track assemblages on Wonthaggi bedding planes.** (A) Bed FF-3, with azimuth orientations between 235–340° for tracks on bedding plane. Depression denoting possible track (PT) also points in northwestern quadrant at 310°; photo scale in centimeters. (B) Bed FF-5, with azimuth orientations between 230–270° for four of five tracks on lower bedding plane (5A) and identical azimuths of 165° for two tracks on upper bedding plane (5B). Ruler (left) = 15 cm long.

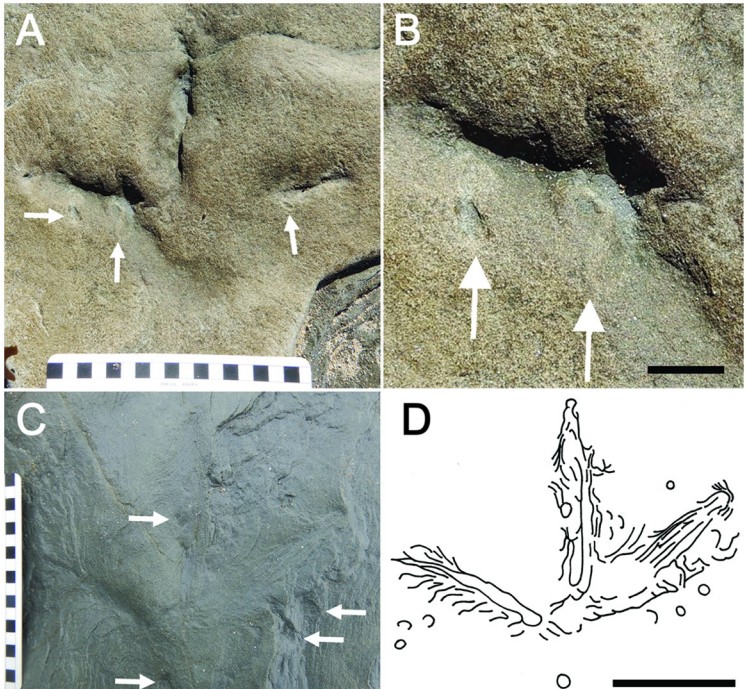

**Fig 16. Invertebrate burrows and avian tracks in Wonthaggi Formation.** (A) Specimen FF-4-2 with oval cross-sections of sand-lined burrows closely associated with track (arrows); scale = 5 mm squares. (B) Close-up of burrow cross-sections on left side of track, with two sand-lined nearly intersecting on right; scale = 1 cm. (C) Specimen FF-5B-4 with circular cross-sections of unlined burrows (arrows); scale = 5 mm squares. (D) Line drawing of FF-5B-4 denoting locations of circular to semi-circular burrow cross-sections with track; scale = 5 cm.

occur with tracks (Fig 16). These burrows are evident as raised, circular to oval outlines inter-secting upper bedding planes; a few had sandy linings (Fig 16B and 16C). None of the burrows are distinctive enough to assign ichnogenera, but they nevertheless indicate infaunal inverte-brates were living in these sediments. As for other vertebrate trace fossils, some strata immedi-ately above and below the studied intervals at both Footprint Flats and Honey Bay also preserve larger non-avian theropod tracks, which are the subject of a separate study by most of the authors.

## Discussion

### Probable identities of trackmakers and sedimentary conditions

All of the described Wonthaggi tracks here are interpreted as those of theropod dinosaurs, but more specifically of avian theropods. Because every track is minimally tridactyl–either aniso-dactyl or anisodactyl incumbent–they readily eliminate a wide range of other potential tetra-pod tracemakers documented from body fossils in the Wonthaggi Formation, such as temnospondyls, chelonians, ceratopsians, ankylosaurs, and mammals [10]. Tridactyl tracks in Cretaceous strata are normally attributed to two broad categories of dinosaurian tracemakers, ornithopods and theropods [46], with theropod tracks further split into non-avian or avian. However, ornithopod tracks typically have equant widths and lengths (length:width ratios about 1.0), thick digits relative to track lengths, blunt ungual impressions, digit II-IV divarica-tion of 90° or less, and no hallux [47 and references therein]. As a result, we do not regard any of the Wonthaggi tracks at either locality described here as those of ornithopods.

As for distinguishing fossil avian tracks from those of non-avian theropods, Lockley and others [25] listed the following diagnostic criteria for avian footprints: (1) resemblance to modern bird tracks; (2) relatively smaller sizes; (3) indefinite phalangeal pad impressions; (4) wide divarication angles between digits II and IV (110–120˚ or more); (5) a rearward-pointing hallux; (6) thin claws; (7) claws on digits II and IV curving away from digit III; and (8) narrow digits relative to footprint length. This checklist has been amended since [27, 46] and augmented by refined descriptions of modern bird tracks as analogues [45, 48].

Given these guidelines, Wonthaggi Formation tracks described in this study meet most of the diagnostic criteria for an avian identity, and some fulfill all prescribed traits, including a posterior-pointing hallux. Those tracks lacking a hallux are relatively small (with some exceptions), have indefinite phalangeal pads, wide divarication angles, thin claws, and narrow digits relative to track length. More than half of the tracks (n = 15) have claw impressions on digits II and IV that curve posteriorly from a midline defined by digit III. Non-avian theropods are the closest alternative tracemakers, but their tracks are characterized by: (1) larger sizes; (2) definite phalangeal pads; (3) narrower divarication between digits II and IV (typically < 90˚) and hence length:width ratios exceeding 1.0; (4) no hallux; (5) thicker claws; (6) claw impressions of digits II and III turned medially, but that of digit IV is turned laterally; and (7) thicker digits relative to track length [25, 27, 46, 47].

The lack of skin impressions or other fine anatomical details of the Wonthaggi tracks, as well as their common occurrence on rippled bed tops, points toward their original preservation either as undertracks, surface tracks that were shallowly impressed, or tracks that were partially eroded before burial. Digit impressions of some tracks on bedding-plane surfaces, such as those of stratum 1 at Footprint Flats (FF-1-1, FF-1-2, etc.), were also evident primarily as linear disturbances of underlying bedding. This expression implies that sediments were moist and collapsed into footprints after feet were pulled out, similar to results from neoichnological experiments using sediments of varying grain size and water content [49, 50]. For example, Falk and others [50] demonstrated that finer-grained (silt-sand) sediments preserve more details in tracks, and that sediment moisture contents of 6.7–8.8% (by volume) result in thinner and tapered digits that are also less likely to record pad impressions. Most Wonthaggi tracks similarly have thin and tapered digits with rare pad impressions, thus these more likely formed in moist sediments on fluvial floodplains.

The fine-grained sediments of the 7-m thick track-bearing sequence at Footprint Flats, consisting of rippled sandy siltstones and silty sandstones bounded by mudstones, suggest these facies were distal overbank-floodplain deposits formed by low flow regimes, contrasting with high-energy conglomeratic alluvial- and fluvial-channel bedforms [32–34, 39]. Considering the circumpolar setting of these facies, high discharge and flow regimes were likely associated with spring thaws and consequent run-off from upland areas, whereas lesser flows were more typical of late spring or early summer. A modern Arctic fluvial point bar in the Colville River shows the same seasonal dynamics, in which massive discharges follow spring thaws, but summer results in ample exposure of the point bar, enabling tracemaking activities and preservation of their traces [24].

## Ichnotaxonomy

Modern weathering and erosion affected Wonthaggi track preservation from their exposure on marine platforms, a daily attrition inflicted by tides, waves, and rocky-intertidal organisms (Fig 14). Owing to these processes and incomplete or indistinct preservation of foot anatomy, previously established ichnotaxa are not readily applied to most tracks. This cautionary approach is especially justified when the Wonthaggi tracks are compared to better preserved

and far more abundant specimens of avian-attributed ichnotaxa from Cretaceous tracksites outside of Australia, such as those in east Asia and North America [27, 51–58].

With that said, a large number of avian-affiliated ichnogenera could be eliminated on the basis of gross morphology. For example, none of the tracks are zygodactyl, nor are any fully palmate with distal webbing, which are forms expressed in Cretaceous bird tracks elsewhere [51, 59–62]. We accordingly evaluated possible ichnogenera by first placing the Wonthaggi specimens into broad anatomical categories of: anisodactyl incumbent with proximal webbing; anisodactyl incumbent without proximal webbing; and anisodactyl without webbing. A few tracks, such as FF-4-1, FF-4-3, and FF-5B-3, still do not correspond with ichnogenera based on known bird-track morphotypes, but the majority are at least comparable to some described outside of Australia (Table 2).

Tracks expressed as anisodactyl incumbent with proximal webbing, such as FF-2-1, FF-3-1, FF-3-5, FF-4-2 and FF-5A-2, somewhat resemble *Gyeongsangornipes* from Early Cretaceous strata of Korea [53, 63, 64], and *Uhangrichnus* from Late Cretaceous strata of Korea and Alaska [65, 66]. However, the Wonthaggi specimens do not exactly match these ichnogenera, as *Gyeongsangornipes* is smaller, has asymmetrical web impressions between traces of digits II-IV, and a higher divarication (~125–130°) between digits II-IV [53, 55, 62]. *Uhangrichnus* is even smaller than the Wonthaggi tracks and has equally developed web impressions between digits II-IV, approaching fully palmate [55, 65]. A better candidate for these Wonthaggi tracks is *Avipeda*, an ichnogenus that is typically applied to Cenozoic avian tracks [21, 62]. For *Avipeda*, tracks are anisodactyl incumbent, digits are shorter and relatively thicker, and webbing is either lacking or limited to the most proximal parts of interdigital angles [67, 68]. However, the Wonthaggi tracks also differ slightly from *Avipeda* in size and divarication. In short, we note that anisodactyl incumbent tracks with proximal webbing are similar to *Avipeda*, but may not entirely fit that ichnogenus, either.

Other Wonthaggi tracks such as FF-3-2, FF-3-3, FF-4-4, FF-4-5, FF-4-7, FF-5A-3 and FF-5A-4, resemble the Cenozoic ichnogenus *Fuscinapeda*, which some researchers have placed in the "ichnofamily" Avipedidae [21, 27, 67, 69]. Such tracks are anisodactyl incumbent and relatively larger than similar ichnogenera, with slim to moderately thick digits, a divarication of about 105°, and webbing either confined to the most proximal parts of interdigital angles or absent [27, 68]. Nonetheless, the Wonthaggi tracks also have a few morphological differences from previously known specimens of *Fuscinapeda*, such as greater size and divarication. Thus we can only suggest that the avian tracemakers of these specimens had ecomorphological affinities to this ichnogenus.

**Table 2. Avian-affiliated ichnogenera that might be applied to Wonthaggi Formation tracks.** See text and S2 File for descriptions, depictions, and diagnoses of individual tracks.

| Ichnogenus | Specimens |
|---|---|
| *Aquatilavipes* | HB-1, FF-1-1, FF-1-2, FF-5A-1 |
| *Ardeipeda* | FF-1-3, FF-1-4 |
| *Avipeda* | FF-2-1, FF-3-1, FF-3-5, FF-4-2, FF-5A-2 |
| *Fuscinapeda* | FF-3-2, FF-3-3, FF-4-4, FF-4-5, FF-4-7, FF-5A-3, FF-5A-4 |
| *Gyeongsangornipes* | FF-2-1, FF-3-1, FF-3-5, FF-4-2, FF-5A-2 |
| *Hwangsanipes* | FF-5B-2, FF-5B-4 |
| *Limiavipes* | 5A-5 |
| *Wupus* | FF-3-6, FF-4-3, FF-4-6, FF-5B-1 |

Wonthaggi tracks that were anisodactyl incumbent but without obvious proximal webbing are similar to a few previously reported Cretaceous avian ichnogenera. Among these are HB-1, FF-1-1, FF-1-2 and FF-5A-1, which align with the ichnogenus *Aquatilavipes* and other ichnogenera from the "ichnofamily" Avipedidae [21, 62, 70–72]. Still, lengths and widths of the Wonthaggi specimens are greater than most *Aquatilavipes*, although size differences in this ichnogenus are reported in other deposits [73]. Wonthaggi track FF-5A-5 is also morphologically similar to the ichnogenus *Limiavipes* ("ichnofamily" Limiavipedidae), but its length and width are slightly smaller than others identified from elsewhere [74]. Some anisodactyl incumbent tracks, such as FF-3-6, FF-4-3, FF-4-6, and FF-5B-1, are probably assignable to the ichnogenus *Wupus* ("ichnofamily" Limiavipedidae), which is likewise an asymmetrical and larger track with a sub-rounded metatarsal impression [21, 27], a feature evident in tracks FF-4-3 and FF-4-5. *Wupus* was originally interpreted as belonging to nonavian theropods, but was later credited to avians [27, 75], affirmed by Lockley and others [21]. Diagnostic features of *Wupus* are its relatively large size (10–11.5 cm long), length:width ratios of 0.7–0.9, digit II-IV divarication angles averaging about 98°, webbing either indistinct or absent, and no hallux [21, 27]. These criteria match at least three of the specimens from Footprint Flats, FF-3-6, FF-4-3, FF-4-6. *Aquatilavipes*, *Limiavipes*, and *Wupus* have been identified in Early Cretaceous strata of eastern Asia and North America, but never before in Gondwanan strata.

Wonthaggi tracks that are anisodactyl (digits I-IV present) but with no webbing evident, such as FF-1-3 and FF-1-4, are similar to the Cenozoic ichnogenus *Ardeipeda*, which resembles tracks of modern long-legged wading birds (e.g., *Ardea alba*, *A. herodias*). Such tracks have three thin forward-pointing digits and a well-developed hallux pointing nearly opposite of those impressions. Ichnospecies reported for this ichnogenus include *A. egretta* and *A. gigantea* [61, 76]. Wonthaggi tracks that are anisodactyl (digits I-IV present) and with proximal webbing, such as FF-5B-2 and FF-5B-4, closely resemble the ichnogenus *Hwangsanipes* ("ichnofamily" Ignotornidae), which also has a long posteromedially oriented hallux and asymmetrical web impressions between digits II-III and III-IV [54, 64, 65]. However, the lengths and widths of FF-5B-2, FF-5B-2, and FF-5B-4 are greater than those of previously known examples of *Hwangsanipes*.

Several Wonthaggi tracks (e.g., FF-1-4) have possible hallux imprints, but distorted sediment in the rear part of the track obscures its presence. In a similar vein, sedimentary structures formed as parts of footprints, such as cohesive mounds and wedges of sediment around track interiors, may result in false proximal or distal webbing between digits [77]. With this factor in mind, we cannot reliably interpret proximal webbing between digits II-III or III-IV, or semipalmated feet, from the Wonthaggi tracks. Thus we prefer to suggest that a few tracks show evidence of proximal webbing, which would accordingly affect their naming.

Ichnotaxonomic conventions aside, the Wonthaggi avian-track assemblage represent an unprecedented variety and number of Early Cretaceous avian tracks from the Southern Hemisphere. Although ichnodiversity does not necessarily equal biodiversity, the morphological and size variations of the Wonthaggi tracks (Fig 13) imply they were made by multiple species of birds during the Early Cretaceous in this part of Australia. Moreover, some of the Wonthaggi specimens, such as FF-4-3, FF-4-6, and FF-4-7, are among the largest known avian tracks reported from the Early Cretaceous [21]. Although some ichnogenera identified here are only "best fits" for now, and the naming of new ichnotaxa may be warranted in the future, we must emphasize that such nomenclatural quibbles will not change our base interpretation of the avian diversity represented by these footprints in this time and place.

## Paleontological and paleoecological significance of the Wonthaggi tracks

The only undoubted avian body fossils from the Wonthaggi Formation include a flight feather [11] and a furcula; the latter was identified as that of an enantiornithine [8]. However, most Wonthaggi tracks more closely resemble those of ornithurines, an identity Martin and others [9] suggested for two bird tracks from the Eumeralla Formation at Dinosaur Cove. The greater age of the Wonthaggi tracks (Barremian-Aptian) versus the Eumeralla tracks (Albian) nonetheless implies the former are more likely from enantiornithean birds. The oldest body fossils of ornithurines are from China (*Archaeornithura*, *Protopteryx*), which are Hauterivian, whereas the oldest enantiornithine other than *Archaeopteryx* is *Noguerornis* from the earliest Barremian of Spain, placing these taxa at about 131–129 *Ma* [78–80]. As mentioned previously, the oldest named enantiornithine in Gondwana is *Cratoavis cearensis* from the Crato Formation (Aptian) of Brazil [12, 13]. The greater age of the Wonthaggi tracks would thus extend the probable record of enantiornithines to earlier in the Cretaceous, and closer to the oldest known birds outside of Gondwana.

This variety of avian tracks in formerly circumpolar Early Cretaceous environments is also paleoecologically significant, suggesting a diverse suite of birds living in these ecosystems between 120–128 *Ma*. Two avian footprints from the geologically younger (Albian) Eumeralla Formation at Dinosaur Cove, dating at about 106 *Ma* [9], further argue for an avian presence in circumpolar ecosystems of Australia for about 15–20 million years. Circumpolar avian tracks are documented in the Northern Hemisphere from Alaska [66, 81], but these are also geologically younger (Campanian-Maastrichtian) than both the Eumeralla and Wonthaggi tracks. In short, the Wonthaggi tracks not only represent the oldest documented bird-track assemblage in Australia and Gondwana, but also the oldest from formerly polar environments.

This presence of birds on rippled sandy floodplain surfaces–presumably during a circumpolar spring or summer–also prompts a query of why they were there. Small invertebrate burrows in the same strata preserving avian tracks provide one answer (Fig 16), as this infauna would have provided food for birds foraging in these environments. Such behaviors have been interpreted from bird tracks and feeding sign in Early Cretaceous rocks of Korea [82]. Bird foraging as flocks on floodplains also may have resulted in similarly oriented tracks, such as those measured on strata 3 and 5 (Fig 15). Furthermore, some tracks, such as FF-5A-3, FF-5A-4, FF-5B-2, and FF-5B-4, have prominent sedimentary structures that imply sudden stops, perhaps indicating landing from either a hop or flight. Flighted behavior was proposed for producing similar features in one of the two avian footprints from the Early Cretaceous Eumeralla Formation [9], and from Early Cretaceous bird tracks of Korea [82].

Regardless of paleolatitude, bird trace fossils are either rare or rarely recognized from Cretaceous strata of former Gondwanan landmasses, and are not yet verified from the Late Jurassic, either. Other than tracks from the Eumeralla Formation (Albian) of Australia [9] and this study, Gondwanan bird tracks have only been reported from Late Cretaceous strata, such as in the Anacleto Formation (Campanian) and Yacoraite Formation (Maastrichtian) of Argentina [14–17], as well as in the Zebbag Formation (Cenomanian) of Tunisia [18]. Late Triassic-Early Jurassic avian-like footprints were interpreted by Ellenberger [83, 84] in South Africa, as well as by others in Argentina [85]. However, the South African tracks are more likely those of non-avian theropods, whereas the Argentine tracks are avian, but were later dated as Late Eocene [86, 87]. As a result, Triassic or Jurassic avian footprints are so far unconfirmed from Gondwanan landmasses.

In contrast to Gondwana, Early and Late Cretaceous bird tracks are abundant and diverse in former Laurasian landmasses, such as North America [62, 66, 70, 73, 81, 88] and Asia [27, 28, 51, 52, 54–58, 71, 89, 90], including those in Early Cretaceous strata [27, 51, 54–58, 60, 72,

89]. The origin of birds is currently hypothesized to have happened during the Middle to Late Jurassic, and in Laurasia [91, 92]. Hence the greater numbers and varieties of Cretaceous bird tracks in Laurasia may reflect proximity to their paleogeographic origin. As of now, we do not know if the uneven distribution of Cretaceous bird tracks in Laurasia and Gondwana is related to: the timing of bird origins and their subsequent evolution and dispersal; preservation bias; researchers not recognizing avian tracks in Gondwanan strata; or a combination of these factors. We nevertheless hope the results of our study provides impetus for researchers to seek more evidence of Early Cretaceous birds in former Gondwanan landmasses, and in particular via their trace fossils.

A paleoecologically significant facet of the Wonthaggi Formation bird tracks is their presence in formerly circumpolar environments during the Early Cretaceous. Their occurrence on fluvial floodplains also implies seasonally linked behaviors, in which they would have walked across emergent surfaces after flooding from spring thaws. Martin [24] similarly concluded that modern avian tracks and other vertebrate tracks on an Arctic point bar were more likely made and preserved during summer, rather than other seasons. The recurrence of avian tracks on at least five stratigraphic levels at Footprint Flats also hints at multiple seasonal behaviors that allowed birds to gather on exposed fluvial floodplains, which also aided in the formation and preservation of their tracks.

The wider implications of a diverse Early Cretaceous avifauna in circumpolar Australia, and flighted birds in particular, may also provide an early glimpse of seasonally linked migrations in birds. In this scenario, Early Cretaceous birds might have flown to Australia from nearby northern regions of Gondwana during Southern Hemisphere springs, then returned to those places during Australia-Antarctica winters. In contrast, paleogeographic maps indicate lengthy oceanic distances between Asia and circumpolar Australia during the Barremian-Aptian [93], which argue against birds having direct north-south migration routes between Laurasia and Gondwana. Nevertheless, the morphological similarity of some Wonthaggi Formation bird tracks to those reported from eastern Asia [58, 88, 90] may also reflect the dispersal of Early Cretaceous flighted birds from Laurasia into other parts of Gondwana before they arrived in Australia. Current assessments of how winged migrations evolved in birds are based on modern range maps and climatic niches viewed through phylogenetic trees [94], but do not include trace-fossil data. In this respect, then, avian trace fossils in polar environments of Early Cretaceous Australia provide important insights on the global paleobiogeography and adaptations of birds relatively early in their evolutionary history.

In summary, through a combination of avian body fossil evidence in the Wonthaggi Formation [8, 10, 11] and trace fossils in the Wonthaggi and Eumeralla Formations [9], we propose that a variety of birds were participating members of polar-ecosystem communities in Australia for at least 15–20 million years during the Early Cretaceous. We anticipate that this ichnologically aided revelation will contribute to and spur on further inquiries about the global dispersal of birds since their origin from non-avian theropods during the Jurassic [91, 92].

## Conclusions

- Tracks in the Wonthaggi Formation (Barremian-Aptian, 128–120 *Ma*) of Victoria, Australia are the earliest known trace fossils attributed to birds in Australia, while also supplementing rare body fossil evidence of birds in the same formation.

- The Wonthaggi avian tracks are also the oldest documented from former Gondwanan landmasses and the oldest from formerly polar environments.

- Based on track forms and sizes, the avian trackmakers were varied, including birds larger than most other Early Cretaceous examples, and with sizes overlapping those of non-avian theropods.

- The Wonthaggi avian tracks occurring on multiple stratigraphic levels indicate a variety of birds on circumpolar floodplains of southern Australia more than 120 *Ma*, and their recurrent presence in those environments suggest seasonal formation of these traces during polar summers, which may also reflect migrations.

## Supporting information

**S1 File. Description of molding and casting process for avian tracks in Wonthaggi Formation.**
(DOCX)

**S2 File. Detailed descriptions of individual tracks from the Wonthaggi Formation augmented by labeled photos and line drawings.**
(PDF)

## Acknowledgments

The research was conducted on the ancestral lands of the Bunurong people, and we appreciate the guidance of Dan Turnbull and the Bunurong team in advising us of respectful practices for studying the paleontological legacy of these places. We are grateful to the Royal Auto Club of Victoria (RACV) for providing discounted lodging for most of us during field work there in May 2022. Our sincere thanks go to Rangers Brian Martin and Gerard Delaney of Parks Victoria for assistance with permits, which made it possible for us to explore the Victorian Cretaceous coast. Partial support for travel and field expenses was provided by our respective institutions. Field work was assisted by Aidan Lowery, Ruth Schowalter, Doris Seeget-Villers, Lesley Kool, and Mike Cleeland. Sally Rogers-Davidson masterfully drafted the colorful Footprint Flats stratigraphic section depicted in Fig 2. We thank Associate Editor Miquel Vall-llosera Camps and two anonymous reviewers for their helpful suggestions for improving the manuscript.

## Author Contributions

**Conceptualization:** Anthony J. Martin, Michael Hall, Patricia Vickers-Rich, Thomas H. Rich.

**Data curation:** Peter Swinkels.

**Formal analysis:** Anthony J. Martin, Claudia I. Serrano-Brañas.

**Investigation:** Anthony J. Martin, Melissa Lowery, Michael Hall, Patricia Vickers-Rich, Thomas H. Rich, Claudia I. Serrano-Brañas.

**Methodology:** Anthony J. Martin, Michael Hall, Claudia I. Serrano-Brañas, Peter Swinkels.

**Visualization:** Anthony J. Martin.

**Writing – original draft:** Anthony J. Martin, Claudia I. Serrano-Brañas.

**Writing – review & editing:** Anthony J. Martin, Michael Hall, Patricia Vickers-Rich, Thomas H. Rich, Claudia I. Serrano-Brañas, Peter Swinkels.

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
