## [Decision Letter · Decision Letter 0]

30 Aug 2023

PONE-D-23-16348Earliest known Gondwanan bird tracks: Wonthaggi Formation (Early Cretaceous), Victoria, AustraliaPLOS ONE

Dear Dr. Martin,

Thank you for submitting your manuscript to PLOS ONE. After careful consideration, we feel that it has merit but does not fully meet PLOS ONE’s publication criteria as it currently stands. Therefore, we invite you to submit a revised version of the manuscript that addresses the points raised during the review process.

I would like to sincerely apologise for the delay you have incurred with your submission. It has been exceptionally difficult to secure reviewers to evaluate your study. We have now received two completed reviews; the comments are available below. The reviewers have raised some scientific concerns about the study that need to be addressed in a revision. 

Please revise the manuscript to address all the reviewer's comments in a point-by-point response in order to ensure it is meeting the journal's publication criteria. Please note that the revised manuscript will need to undergo further review, we thus cannot at this point anticipate the outcome of the evaluation process.

We look forward to receiving your revised manuscript.

Kind regards,

Miquel Vall-llosera Camps

Senior Editor

PLOS ONE

Journal Requirements:

"No authors have competing interests."

4. We note that Figure 1 in your submission contain map/satellite images which may be copyrighted. All PLOS content is published under the Creative Commons Attribution License (CC BY 4.0), which means that the manuscript, images, and Supporting Information files will be freely available online, and any third party is permitted to access, download, copy, distribute, and use these materials in any way, even commercially, with proper attribution. For these reasons, we cannot publish previously copyrighted maps or satellite images created using proprietary data, such as Google software (Google Maps, Street View, and Earth). For more information, see our copyright guidelines: http://journals.plos.org/plosone/s/licenses-and-copyright.

Reviewers' comments:

Reviewer's Responses to Questions

**Comments to the Author**

1. Is the manuscript technically sound, and do the data support the conclusions?

Reviewer #1: Yes

Reviewer #2: Yes

2. Has the statistical analysis been performed appropriately and rigorously? 

Reviewer #1: Yes

Reviewer #2: Yes

3. Have the authors made all data underlying the findings in their manuscript fully available?

Reviewer #1: Yes

Reviewer #2: Yes

4. Is the manuscript presented in an intelligible fashion and written in standard English?

Reviewer #1: Yes

Reviewer #2: Yes

5. Review Comments to the Author

Reviewer #1: I have very little to comment on regarding the actual text of the paper, so I will submit my comments here. The manuscript details a novel avian ichnofauna from Early Cretaceous Gondwana. In short, this is a tremendous find, and provides important information on the paleodiversity of early avians. Given the challenging nature of the present-day environment and the nature of the preservation, the authors approached the description of these tracks with an appropriate level of caution: suggestions of possible ichnotaxonomic assignments were made without formal designation. This provides the opportunity for more detailed examination of the tracks and the possibility for future surveys of the Footprint Flats and Honey Bay localities for more specimens.

Detailed measurements and data tables were provided, which will be very useful for future analyses, and the images in the paper and the supplemental information were clear and without overinterpretation. I agree with the authors that photogrammetric images are not required at this stage. Also, the encrusting tidal fauna on many of the tracks would provide a great deal of artificial relief to a heat map.

The only item that I would like to see expanded on may be the possible migratory potential for volant Early Cretaceous avians to polar regions during the spring and summer as we see in many of our modern analogs. Would a north-south migration of Barremian – Aptian shore and wading birds during the breeding season have any geographic barriers?

I recommend that PLOS One accept this paper for publication.

Reviewer #2: A competent paper with broad coverage of avian tracks literature

I am a little surprised to see the Wonthaggi tracks compared with such a wide variety of ichnogenera. it would be nice to narrow it down a bit if possible. For example, tracks are much larger the Geongsangornipes and lack web traces.

RE morphometric tables it would be nice to place a line with mean measurements at bottom of each table

Please refer to digit traces not "actual" digits when appropriate

6. PLOS authors have the option to publish the peer review history of their article (what does this mean?). If published, this will include your full peer review and any attached files.

Reviewer #1: No

Reviewer #2: No

---

## [Author Response · Author response to Decision Letter 0]

21 Sep 2023

The uploaded response letter contains all of our responses to the reviewer and editor comments. However, if that information needs repeating here, it is copied and pasted below.

Dear Dr. Miquel Vall-llosera Camps,

Thank you very much for the helpful reviews of our research article, Earliest known Gondwanan bird tracks: Wonthaggi Formation (Early Cretaceous), Victoria, Australia (PONE-D-23-16348), which we originally submitted to PLoS One on May 27, 2023. My coauthors and I are satisfied that the reviews were both thorough and fair, and in this letter, we address any concerns or suggestions provided by the two external reviewers. 

Reviewer #1 (Reviewer concerns and suggestions in italics, our replies in bold.)

I have very little to comment on regarding the actual text of the paper, so I will submit my comments here. The manuscript details a novel avian ichnofauna from Early Cretaceous Gondwana. In short, this is a tremendous find, and provides important information on the paleodiversity of early avians. Given the challenging nature of the present-day environment and the nature of the preservation, the authors approached the description of these tracks with an appropriate level of caution: suggestions of possible ichnotaxonomic assignments were made without formal designation. This provides the opportunity for more detailed examination of the tracks and the possibility for future surveys of the Footprint Flats and Honey Bay localities for more specimens.

We appreciate Reviewer #1’s affirmation of the paleontological importance of these trace fossils, which we hope will expand our understanding of Early Cretaceous avian biodiversity and paleobiogeography. We also agree that our study represents an opportunity for further study of these tracks, while encouraging further field work aimed at discovering and documenting more specimens in coastal exposures of the Wonthaggi Formation near Inverloch, Victoria.

Detailed measurements and data tables were provided, which will be very useful for future analyses, and the images in the paper and the supplemental information were clear and without overinterpretation. I agree with the authors that photogrammetric images are not required at this stage. Also, the encrusting tidal fauna on many of the tracks would provide a great deal of artificial relief to a heat map.

We agree with the reviewer that our descriptions and interpretations should contain enough evidence for future workers interested in re-studying these Early Cretaceous bird tracks, and that our methods for studying the tracks were appropriate given their modern environmental setting and preservational conditions.

The only item that I would like to see expanded on may be the possible migratory potential for volant Early Cretaceous avians to polar regions during the spring and summer as we see in many of our modern analogs. Would a north-south migration of Barremian–Aptian shore and wading birds during the breeding season have any geographic barriers?

We thank the reviewer for suggesting that we link our reported avian footprints with the migratory potential of Barremian-Aptian flighted birds. As a result, we have added the following paragraph on that topic in the manuscript, as well as two new references cited in that paragraph.

The wider implications of a diverse Early Cretaceous avifauna in circumpolar Australia, and flighted birds in particular, may also provide an early glimpse of seasonally linked migrations in birds. In this scenario, Early Cretaceous birds might have flown to Australia from nearby northern regions of Gondwana during Southern Hemisphere springs, then returned to those places during Australia-Antarctica winters. In contrast, paleogeographic maps indicate lengthy oceanic distances between Asia and circumpolar Australia during the Barremian-Aptian [95], which argue against birds having direct north-south migration routes between Laurasia and Gondwana. Nevertheless, the morphological similarity of some Wonthaggi Formation bird tracks to those reported from eastern Asia [90-92] may also reflect the dispersal of Early Cretaceous flighted birds from Laurasia into other parts of Gondwana before they arrived in Australia. Current assessments of how winged migrations evolved in birds are based on modern range maps and climatic niches viewed through phylogenetic trees [96], but do not include trace-fossil data. In this respect, then, avian trace fossils in polar environments of Early Cretaceous Australia provide important insights on the global paleobiogeography and adaptations of birds relatively early in their evolutionary history.

References:

95. Scotese CR. Atlas of Early Cretaceous Paleogeographic Maps 2014; PALEOMAP

Atlas for ArcGIS, 2, The Cretaceous, Maps 23-31, Mollweide Projection, PALEOMAP Project, Evanston, IL.

96. Dufour P, Descamps S, Chantepie S, Renaud, J, Guéguen M, Schiffers K, Thuiller W, Lavergne S. Reconstructing the geographic and climatic origins of long-distance bird migrations. Journal of Biogeography 2020;47: 155-166.

Reviewer #2 (Reviewer concerns and suggestions in italics, our replies in bold.)

Reviewer #2: A competent paper with broad coverage of avian tracks literature

I am a little surprised to see the Wonthaggi tracks compared with such a wide variety of ichnogenera. it would be nice to narrow it down a bit if possible. For example, tracks are much larger the Geongsangornipes and lack web traces.

We justify our comparison of the Wonthaggi Formation tracks to a wide variety of ichnogenera from elsewhere because these tracks represent the oldest documented avian tracks in Australia and the remainder of the Southern Hemisphere. Moreover, the tracks reflect a previously unknown diversity of track forms and sizes from Early Cretaceous strata of those places. Thus our comprehensive coverage of avian-affiliated ichnogenera was intended to benefit researchers who can then more easily compare ichnogenera from widely separated places, such as those from eastern Asia and North America. We also meant to provide future researchers with a checklist of avian ichnogenera (e.g., Table 2) that might prove useful when documenting future discoveries of other Cretaceous avian footprints, whether in Australia or elsewhere.

In short, we advocate for erring on the side of comparing “too many” avian-affiliated ichnogenera rather than “too few,” and do not feel a need to narrow down the list of ichnogenera in our report.

RE morphometric tables it would be nice to place a line with mean measurements at bottom of each table.

This is a good suggestion, as it will aid readers when comparing overall statistics (Table 1D) to those of anisodactyl incumbent and anisodactyl tracks (Table 1A and 1B, respectively). As a result, we added mean and standard deviation at the bottom of each of the first two parts of Table 1. However, we did not add these values for Table 1C, which only had measurements for two tracks and included extrapolated widths and length:width ratios.

Please refer to digit traces not "actual" digits when appropriate.

We included most of Reviewer #2’s suggested editorial changes in the manuscript, such as adding “trace” or “traces of” to “digit” and other anatomical terms related to foot morphology of the tracks. However, we did not change “active marine platform” to the reviewer’s preferred “marine platform setting” or “wave-cut platform setting,” which we felt unnecessarily complicated its description. Similarly, we meant to use “sketches” rather than “tracings” (p. 16) when describing our initial field assessments of the tracks, so this is unchanged.

Associate Editor (editor concerns and suggestions in italics, our replies in bold)

The resubmitted manuscript differs from the original in the following respects:

- Added a paragraph suggested by Reviewer #1 on possible connections between Early Cretaceous bird tracks in a circumpolar region of Gondwana and winged migrations.

- Added two references cited in that paragraph to the bibliography.

- Made most of the minor editorial corrections in the manuscript suggested by Reviewer #2.

- Made further minor editorial corrections to the final manuscript by one of us (AJM) after proofreading (shown in the tracked changes).

- Added to the Acknowledgements, with three of us (PVR, AJM, MH) contributing to that.

- Added a list of tracks in this study that were molded and cast in Supporting Information 2.

- Uploaded, corrected, and saved all figures using the Preflight Analysis and Conversion Engine (PACE) digital diagnostic tool.

We also note that Figure 1B, which is a satellite image of the field area, was downloaded from U.S.G.S. Earth Explorer site (https://earthexplorer.usgs.gov), which is among the approved public-domain sources listed by PLoS One for Creative Commons (CC BY 4.0). However, if this figure requires further sourcing or replacing, we would appreciate specific directions on how to comply.

As for competing interests, the authors again declare that no competing interests exist.

Thank you again for your consideration of our manuscript, and please let us know if you require anything more from us to satisfy the publication requirements of PLoS One.

Sincerely,

Anthony J. Martin

Department of Environmental Sciences

Emory University

---

## [Editor Report · Decision Letter 1]

10 Oct 2023

Earliest known Gondwanan bird tracks: Wonthaggi Formation (Early Cretaceous), Victoria, Australia

PONE-D-23-16348R1

Dear Dr. Martin,

We’re pleased to inform you that your manuscript has been judged scientifically suitable for publication and will be formally accepted for publication once it meets all outstanding technical requirements.

Kind regards,

Jun Liu

Academic Editor

PLOS ONE
---

## [Editor Report · Acceptance letter]

23 Oct 2023

PONE-D-23-16348R1 

Earliest known Gondwanan bird tracks: Wonthaggi Formation (Early Cretaceous), Victoria, Australia 

Dear Dr. Martin:

I'm pleased to inform you that your manuscript has been deemed suitable for publication in PLOS ONE. Congratulations! Your manuscript is now with our production department. 

Kind regards, 

on behalf of

Dr. Jun Liu 

Academic Editor

PLOS ONE